# Frequency and Associated Factors of Suicidal Ideation in Patients with Chronic Obstructive Pulmonary Disease

**DOI:** 10.3390/jcm11092558

**Published:** 2022-05-02

**Authors:** Carlos Roncero, Joselín Pérez, Jesús Molina, José Antonio Quintano, Ana Isabel Campuzano, Javier Pérez, Marc Miravitlles

**Affiliations:** 1Psychiatric Service, University of Salamanca Health Care Complex, 37007 Salamanca, Spain; jperezro@saludcastillayleon.es; 2Institute of Biomedicine of Salamanca (IBSAL), University of Salamanca, 37007 Salamanca, Spain; 3Medical Department, Grupo Ferrer, 08029 Barcelona, Spain; mjperez@ferrer.com (J.P.); acampuzano@ferrer.com (A.I.C.); 4Centro de Salud Francia, Dirección Asistencial Oeste, 28993 Madrid, Spain; jmolinaparis@gmail.com; 5Centro de Salud Lucena I, Lucena, 14900 Córdoba, Spain; quintanojimenez@gmail.com; 6Department of Pneumology, Vall d’Hebron Institut de Recerca (VHIR), Hospital Universitari Vall d’Hebron, Vall d’Hebron Barcelona Hospital Campus, 08035 Barcelona, Spain; 7Centro de Investigación Biomédica en Red de Enfermedades Respiratorias (CIBERES), 28029 Madrid, Spain

**Keywords:** depression, suicide, COPD, Beck Depression Inventory

## Abstract

We aimed to examine the prevalence of suicidal ideation in patients with chronic obstructive pulmonary disease (COPD) and the association between demographic and clinical variables and the occurrence of suicidal thoughts. This was a cross-sectional study. Sociodemographic and clinical data were recorded, and questionnaires were used to assess depressive symptoms (Beck Depression Inventory), comorbidities (Charlson Index), cognitive performance (Mini Mental State Examination), and quality of life (EuroQoL-5 dimensions and CAT). Specific questions about suicide-related behavior were included. Multivariate logistic regression analysis identified the significant factors associated with previous suicidal ideation and suicide attempts. The analysis included 1190 subjects. The prevalence of suicidal ideation and suicide attempts were 12.1% and 2.5%, respectively. Severely depressed patients had the highest prevalence of suicide-related behavior. The adjusted logistic model identified factors significantly associated with suicidal ideation: sex (odds ratio (OR) for women vs. men = 2.722 (95% confidence interval (CI) = 1.771–4.183)), depression score (OR = 1.163 (95% IC = 1.127–1.200)), and Charlson Index (OR 1.228 (95% IC 1.082–1.394)). Suicidal ideation is common in COPD patients, especially in women. While addressing suicidal ideation and suicide prevention, clinicians should first consider the management of depressive symptomatology and the improvement of coping strategies.

## 1. Introduction

Chronic obstructive pulmonary disease (COPD) is a chronic and progressive respiratory disease that is frequently associated with multiple comorbidities [1]. Among these comorbidities, depression is one of the most frequent, with a prevalence of a clinical diagnosis ranging between 10% and 42% [2,3]. There is not a clear picture of all the associated factors with suicide ideation among COPD patients, but personality traits [4] and depression status [5] may be among them. It has also been suggested that the distress of suffering a severe physical disorder, the functional limitations, and the feelings of perceived burdensomeness may increase the risk of suicidal behaviors in older adults [6], and in fact, patients with severe and chronic physical disease carry a risk for committing suicide [7]. Few studies have examined the link between COPD and suicidality, but their results have consistently indicated an association between COPD and elevated suicide risk [8,9,10].

An optimal strategy for addressing psychiatric comorbidities is to appreciate the heightened risk in this specific clinical population and to recognize the risk factors for suicide, such as a history of self-harm [11]. However, it is not always easy for patients to spontaneously report self-injurious/suicidal behaviors, nor can their treating physicians make determinations by simply observing the patient’s mood. The early identification of risk factors that could be associated with suicide-related behaviors may assist healthcare professionals to make appropriate treatment and referral decisions, with the final aim of preventing suicides among COPD patients. Therefore, the primary objective of this research was to examine the prevalence of suicidal ideation and suicide attempts in an unselected and representative group of patients with COPD; the secondary objective was to identify which major demographic and clinical variables were significantly associated factors, such as sex, depression, and quality of life.

## 2. Methods

### 2.1. Study Design and Sample

This was a cross-sectional and observational study (DeprEPOC or study of Depression in COPD patients). The design of the study has been described in detail previously [5,12]. Briefly, patients were included from primary care centers and pneumology services in Spain. Patients were included if they were 40 years of age or older, smokers or ex-smokers of at least 10 pack-years and had stable COPD (confirmed by post-bronchodilator spirometry showing FEV1/FVC <0.70 and absence of exacerbations for at least 3 months) [13]. All patients that correctly completed the Beck Depression Inventory (BDI) questionnaire and answered the questions about previous suicidal behaviors were included in this post-hoc analysis. The study was approved by the Ethics Committee of the Barcelona Clinic Hospital (Barcelona, Spain) and was conducted in accordance with the principles of the Declaration of Helsinki. All patients provided signed informed consent prior to their participation in the study.

### 2.2. Study Assessments

Information about patients’ sociodemographics and clinical data were collected through face-to-face interviews with the patients and from medical records. Severity of respiratory disease was assessed with the modified Medical Research Council dyspnea scale (mMRC) [14] and the BODEx index (Body mass index, Obstruction, Dyspnea and Exacerbations) [15], the Charlson Index was used to quantify comorbidities [16], and information was obtained about the exacerbations suffered in the previous year [17]. The short Beck Depression Inventory (BDI) questionnaire was used to assess and quantify depressive symptoms [18]. The BDI is a 13-item self-administered questionnaire that assesses affective, motivational, cognitive, and vegetative symptoms of depression. Specific questions about suicidal ideation and suicide attempts were included: “Have you ever thought about ending your life?” and “Did you ever try to end your life?”

Cognitive status was evaluated by means of the Mini Mental State Examination (MMSE) [19], whereas quality of life was assessed by the generic EuroQoL-5 Dimensions (EQ-5D) questionnaire [20] and the specific COPD Assessment Test (CAT) questionnaire [21]. The EQ-5D consists of a five-item descriptive system (including usual activities, self-care, pain/discomfort, mobility, and anxiety/depression), and an overall score or tariff is calculated, ranging from zero (worst) to 100 (best). In addition, a visual analog scale (VAS) that ranges from zero to 100 was scored by the participants, where zero is the worst and 100 is the best health status possible. The validated Spanish version of the CAT questionnaire was used [22]. This is a short, respiratory-specific, quality of life questionnaire for patients with COPD. The CAT consists of eight items, with scores ranging from zero to five, providing a global score out of 40, where zero is the best and 40 the worst state possible. Physical activity was measured by patients’ self-reported average minutes walked per day, as validated by our group in a previous publication [23].

### 2.3. Statistical Analysis

The mean and the standard deviation (SD) were used to describe continuous variables, whereas absolute and relative frequencies were used for categorical variables. No imputation was conducted for missing values. Pairwise comparisons of qualitative variables were performed by the Fisher’s exact test, Bonferroni-corrected. In order to determine the relationship between quantitative variables by group, the Student’s *t*-test was used. Stepwise multivariate logistic regression analysis with adjusted odds ratios (OR) were conducted to evaluate the different risks predefined in the study. In this analysis, we included all clinical, demographic, and questionnaire variables. The first model was developed with suicidal ideation/attempt history as a dependent variable, and independent variables were considered to be all variables that showed a significant association in the bivariate analysis (model 1). Model 2 was constructed with the same methodology but including only demographic and clinical variables among the independent variables, excluding the questionnaires. This model 2 was built with the objective of identifying the variables associated with suicidal ideation/attempt history that can be identified by physicians in everyday clinical practice without the administration of questionnaires. Finally, all multivariate analyses that showed significant results were studied by receiver operating characteristic (ROC) curve analysis.

A *p*-value <0.05 was considered significant. The statistical analysis was conducted with SAS software version 9.1.3 Service Pack 3 (Cary, NC, USA, EE.UU.). The investigators recruiting patients were different from the steering committee that conducted the analysis.

## 3. Results

### 3.1. Patients’ Characteristics and Suicidal Behavior

Of 1273 patients screened by 343 investigators, 1190 were valid for the analysis (Figure 1). Included patients were predominantly male (80.2%), with a median age of 68.0 years (range: 40 to 90 years). A total of 144 patients (12.1%) confirmed the presence of previous suicidal ideation. Of them, 34 (23.6%) had at least one suicide attempt and 10 (29.4%) cases required admission to a psychiatric unit (Figure 1).

As shown in Table 1, suicidal ideation was more frequent in women, in younger patients, in current smokers, and in patients with longer disease duration, more severe dyspnea, and more COPD exacerbations in the previous year. In those receiving domiciliary oxygen, suicidal ideation was also more prevalent. Patients presenting with more depressive symptoms and patients with lower cognitive function more often reported suicidal thoughts, and high comorbidity burden and poor quality of life were also associated with suicidal ideation (Table 1). When patients were categorized into four groups according to BDI score (no depression, mild depression, moderate depression, and severe depression), frequencies of suicidal ideation were 9/301 (2.9%), 13/272 (4.9%), 57/430 (13.3%) and 65/178 (36.5%), respectively (*p* < 0.001).

The description of patients that had and did not have a previous suicide attempt is shown in Table 2. In general, a similar pattern of associations was observed between suicide attempts and clinical and patient-reported measures. In this case, no difference in history of suicide attempts was found between female and male patients. Frequencies of suicide attempts in patients without depression and with mild, moderate or severe depression were 2/298 (0.7%), 6/264 (2.3%), 10/414 (2.4%), and 16/170 (9.4%), respectively (*p* < 0.001).

### 3.2. Factors Associated with Previous Suicidal Behaviors

In the multivariate analysis including all the variables that were significant in the comparative analysis, suicidal ideation was associated with female sex, more depressive symptoms, and higher comorbidity burden (Table 3). The c index (area under the ROC curve) from this multivariable model was 0.81 (confidence interval (CI) 95%, 0.77 to 0.85). With the best discriminating Youden’s Index of 0.16, sensitivity and specificity of the model were 0.65 and 0.85, respectively. When we excluded from the analysis all the variables derived from the questionnaires and included only the demographic and clinical variables available in the routine clinical visit, only sex and degree of dyspnea were significant in the multivariate analysis (Table 3). The c index from this multivariable model was 0.68 (CI 95%, 0.63 to 0.73). With the best discriminating Youden’s Index of 0.17, sensitivity and specificity of the model were 0.52 and 0.73, respectively.

The predictive models for suicide attempts are shown in Table 4. Those variables that were statistically significant in the bivariate analysis were selected for multivariate logistic regression analysis, of which only age and BDI score were independently associated with previous suicide attempt(s). The corresponding values of c index, Youden’s Index, sensitivity, and specificity were 0.80 (CI 95%, 0.72–0.88), 0.019, 0.63, and 0.82. In the second model that included only demographic and clinical variables, the significant factors were age and number of outpatient exacerbations (Table 4). The c index from this multivariable model was 0.71 (CI 95%, 0.61 to 0.81). With the best discriminating Youden’s index of 0.039, sensitivity and specificity of the model were 0.59 and 0.76, respectively.

## 4. Discussion

Up to 11% of adults in Spain have COPD [24], and previous reports about the high incidence of comorbidities, particularly depression and anxiety, have contributed to the growing attention to mental health in this population [25,26]. In this study, we found that 12% of the sample of unselected COPD patients reported previous suicidal ideation, and almost 3% had experienced at least one suicide attempt.

It is recognized that people with COPD are more likely to commit suicide [27]. The frequencies of suicidal ideation and suicide attempts in patients with COPD observed in our study are similar to those reported in previous studies [8,28,29]. According to a large case-control study [8], the relative risk of suicide was significantly elevated among patients with COPD compared with patients without major chronic illnesses (3.1% versus 1.9%, respectively). A national survey conducted in Korea showed even higher numbers, with suicidal thoughts reported by 16.0% of patients in GOLD stages I and II and by 23.8% of those in stages III and IV [30].

Some authors suggested that different types or degrees of suicidal ideation represent different levels of risk for suicide, ranging from passive death wishes, to active thoughts of committing suicide, to having a specific suicidal plan with a real intention to carry it out [31]. Although suicidal ideation is a risk factor for completed suicide, thoughts of death may be common among older adults and can represent normative reflections on mortality [32]. In the USA general population, there are 25 attempts for every death by suicide, whereas in the older population (65+ years) the rate is 1:4 [33]. In our study, the specific questions about suicidal ideation and suicide attempts were intended to search for active thoughts/actions of committing suicide, but patients’ incorrect interpretation cannot be ruled out. However, the fact that suicidal behaviors were more common among severely depressed patients indicates that suicidality was correctly identified in the study population. This finding was also observed in a Chinese study, in which suicidal ideation was significantly associated with the severity of depressive symptoms in COPD patients [34] and in Taiwanese COPD patients, depression was associated with a 13.6 percent higher risk of suicidal attempts compared to patients without depression [35]. In a population with psychiatric disorders, the comorbidity of depression and anxiety was the most important risk factor for suicide attempts [36]. Unfortunately, however, no information regarding anxiety was collected in our study.

Patients with suicidal ideation presented with higher COPD severity according to longer disease duration, higher intensity of dyspnea, more exacerbations in the previous year, and the need for oxygen therapy. They also had a higher comorbidity burden and more severely impaired quality of life, with the same pattern observed in suicide attempters, and all these variables are presumably associated with frequent hospitalizations [37]. It has been shown that having required medical attention for several different physical diseases [38] or having been recently hospitalized [39] is linked to elevated risk of suicidal behaviors. Strid et al. [40] demonstrated that there was a substantially increased risk of suicide among patients previously hospitalized for COPD compared with non-hospitalized patients, and the relative risk of suicide increased with the number of hospitalizations and recentness of the last hospital stay.

The identification of suicidal ideation in COPD is particularly important because these patients are candidates for specialized care that can result in improved outcomes. By means of multivariate logistic regression analysis which combined demographic, clinical, and patient questionnaire scores (model 1), female sex, more severe depressive symptoms and high comorbidity levels were significantly and independently associated with the presence of suicidal ideation, with good values of sensitivity and specificity. As the use of questionnaires is not a routine practice by most primary care physicians or even respiratory specialists, we explored an alternative multivariate model including only demographic and clinical variables (model 2). In this model, suicidal ideation was associated with female sex and more severe dyspnea, but the sensitivity and the specificity of the model were lower.

On the other hand, the results for the multivariate associations with suicide attempts showed that age and depression were the only significant risk factors, with good predictive accuracy. After excluding the questionnaires, suicide attempts were associated with age and higher number of exacerbations, but the model lost sensitivity. The high specificity and low sensitivity found using the combination of clinical and demographic variables suggests that their utility may be in confirmatory testing among already selected high-risk individuals, rather than for initial screening.

Our results concur with previous investigations that reported more frequent suicidal ideation in women and smokers [25,41,42]. Active smokers also had more suicidal ideation and suicide attempts than former smokers, which is in line with several reports on independent associations between suicide attempts and nicotine dependence [43] and tobacco use [44,45], and between completed suicide and smoking [46]. Decreased brain serotonin synthesis associated with the hypoxia present in COPD patients, and aggravation in active smokers may be a possible underlying mechanism of the elevated risk of suicide-related behaviors in these populations [47], somehow supported by the finding that suicide rates are also elevated among people living at higher altitudes [48]. However, in our study, active smoking was no longer significant in multivariate analysis.

COPD is a chronic, progressive and debilitating disease; the functional decline that leads to increased dependence on help [49] and the disconnection from social networks interferes with quality of life as well as adherence to medication [50]. During the clinical consultation, asking COPD patients about how they are getting on with managing their medicines and activities needed for their medical condition may provide clues to these patients’ suicidal thoughts. Effective prevention of suicide requires a multifaceted approach in both respiratory medicine and primary care settings, targeting depression and indicators of social support. Involving family members in the development and implementation of treatment plans is essential [51,52]. Patients with high risk should be referred to intensive programs from mental health resources [53].

Our study has some limitations; firstly, the cross-sectional design of this study precludes any type of causal inferences, and the directionality between COPD or its severity and suicidal behavior remains unclear. Life stressors can impact both tobacco use and suicidality, but the social, emotional, and medical contexts of suicidal ideation episodes and suicide attempts were not investigated. Secondly, since conducting non-intervention studies that may influence prescription is prohibited by Spanish legislation, we did not collect any information about prescribing habits to make sure that no interference existed. As treatment may impact the mental health of COPD patients [54], such information should be available in the future. Thirdly, a large multicenter study with the participation of many investigators is subjected to the possibility of missing values in some variables, as occurred in our study. In any case, our results in a large sample of unselected COPD patients confirm previous evidence indicating that chronic medical illness resulting in functional disability may be associated with an increased risk of suicidal thoughts and suicidal attempts even in the absence of previous mental illness [55].

## 5. Conclusions

Suicidal ideation was not uncommon in COPD and was associated with female sex, more severe depressive symptoms, active smoking, more frequent exacerbations, increased comorbidity burden, and low health status. Health professionals should be aware of and discuss suicidality with their patients, as they may need a more individualized intervention to help them cope with their chronic illness. COPD patients who show any signs of depression or distress should be asked about psychological symptoms, including suicidal ideation.

## Figures and Tables

**Figure 1 jcm-11-02558-f001:**
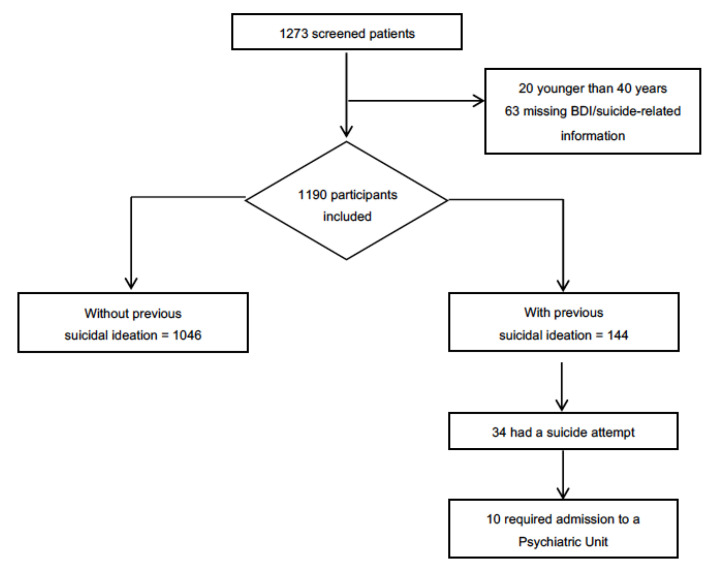
Flow diagram for the analysis and sample distribution according to previous suicidal ideation.

**Table 1 jcm-11-02558-t001:** Comparison of patients with and without previous suicidal ideation.

	“Have You Ever Thought about Ending Your Life?”		
Variable	No (*n* = 1046)	Yes (*n* = 144)	*p*
Sex (*n*, %)			<0.001
Males	854 (82.2)	94 (65.7)	
Females	185 (17.8)	49 (34.3)	
Age (mean, SD)	68.0 (9.5)	65.7 (9.4)	0.007
Coexistence (*n*, %)			0.002
Alone	136 (13.0)	24 (16.9)	
With partner	736 (70.6)	78 (54.9)	
With family	146 (14.0)	34 (23.9)	
Institutionalized	9 (0.9)	3 (2.1)	
With professional caregiver	16 (1.5)	3 (2.1)	
Education level (*n*, %)			0.002
Basic literacy	87 (8.4)	27 (19.0)	
Primary level	619 (59.6)	75 (52.8)	
Secondary level	215 (20.7)	22 (15.5)	
Completed university	117 (11.3)	18 (12.7)	
Smoking habit (*n*, %)			0.013
Ex-smoker	792 (78.3)	95 (68.3)	
Active smoker	220 (21.7)	44 (31.7)	
Pack-years (mean, SD)	39.5 (21.6)	36.9 (22.3)	0.194
COPD duration (mean, SD)	11.1 (7.1)	12.9 (9.3)	0.009
Spirometry (mean, SD)			
FVC, mL	3028 (981)	3001 (933)	0.769
FVC, %	70.4 (17.7)	66.9 (20.4)	0.085
FEV1, mL	1864 (845)	1873 (856)	0.914
FEV1, %	55.2 (17.9)	53.3 (31.6)	0.346
FEV1/FVC	58.3 (20.7)	55.4 (15.1)	0.125
mMRC score (mean, SD)	2.76 (0.89)	3.29 (1.03)	<0.001
Exacerbations in the previous year (*n*, % )			0.004
Yes	881 (84.2)	134 (93.1)	
No	165 (15.8)	10 (6.9)	
Number of exacerbations in the previous year (mean, SD)	3.9 (3.3)	5.0 (4.8)	< 0.001
Number of ambulatory exacerbations (mean, SD)	2.6 (1.9)	3.2 (2.8)	<0.001
Number of admissions (mean, SD)	1.3 (2.0)	1.8 (2.4)	0.014
Home oxygen therapy (*n*, %)			0.002
Yes	211 (21.6)	48 (33.8)	
No	765 (78.4)	94 (66.2)	
BODEx (mean, SD)	2.7 (1.8)	3.6 (2.0)	<0.001
BDI score (mean, SD)	8.1 (5.5)	16.0 (8.1)	<0.001
MMSE (mean, SD)	26.7 (3.8)	23.7 (5.1)	<0.001
Charlson Index (mean, SD)	1.3 (1.3)	2.5 (2.1)	<0.001
CAT score (mean, SD)	20.9 (8.1)	26.5 (7.6)	<0.001
EQ-5D VAS (mean, SD)	58.9 (18.1)	47.1 (19.6)	<0.001
EQ-5D Tariff (mean, SD)	64.8 (23.2)	41.1 (24.8)	<0.001
Minutes walked per day (mean, SD)	68.6 (58.2)	48.2 (39.4)	<0.001

Totals may not add up to the total number of patients due to missing values. Percentages are calculated without missing values. Comparisons were performed with the Fischer’s exact test for qualitative variables and the Student’s *t*-test for quantitative variables, Bonferroni-corrected. Abbreviations: SD, standard deviation; COPD, chronic obstructive pulmonary disease; BODEx, Body mass index, airflow Obstruction, Dyspnea and Exacerbations; BDI, Beck Depression Inventory; MMSE, Mini Mental State Examination; CAT, COPD Assessment Test; EQ-5D, EuroQoL-5 dimensions; VAS, visual analogic scale.

**Table 2 jcm-11-02558-t002:** Comparison of patients with and without previous suicide attempts.

	“Did You Ever Try to End Your Life?”		
Variable	No (*n* = 1112)	Yes (*n* = 34)	*p*
Sex (*n*, %)			0.261
Males	894 (80.9)	24 (72.7)	
Females	211 (19.1)	9 (27.3)	
Age (mean, SD)	67.9 (9.5)	63.2 (9.5)	0.005
Coexistence (*n*, %)			0.249
Alone	147 (13.6)	7 (21.2)	
With partner	772 (69.6)	19 (57.6)	
With family	165 (14.9)	6 (18.2)	
Institutionalized	10 (0.9)	1 (3.0)	
With professional caregiver	15 (1.3)	0 (0)	
Education level (*n*, %)			0.594
Basic literacy	106 (9.6)	5 (15.1)	
Primary level	649 (58.8)	19 (57.6)	
Secondary level	222 (20.1)	7 (21.2)	
Completed university	126 (11.4)	2 (6.1)	
Smoking habit (*n*, %)			0.010
Ex-smoker	843 (78.0)	19 (57.6)	
Active smoker	238 (22.0)	14 (42.4)	
Pack-years (mean, SD)	38.9 (21.6)	43.3 (22.3)	0.263
COPD duration (mean, SD)	11.4 (7.5)	10.7 (7.2)	0.566
Spirometry (mean, SD)			
FVC, mL	3014 (972)	3047 (890)	0.863
FVC, %	70.0 (18.1)	73.5 (15.7)	0.397
FEV1, mL	1852 (837)	2000 (886)	0.356
FEV1, %	54.7 (18.1)	57.7 (22.3)	0.471
FEV1/FVC	57.8 (20.4)	60.1 (15.9)	0.556
mMRC score (mean, SD)	2.81 (0.91)	3.03 (1.00)	0.177
Exacerbations in the previous year (*n*, %)			0.214
Yes	947 (85.2)	32 (94.1)	
No	165 (14.8)	2 (5.9)	
Number of exacerbations in the previous year (mean, SD)	4.0 (3.5)	6.0 (5.1)	0.001
Number of ambulatory exacerbations (mean, SD)	2.6 (2.0)	3.9 (3.6)	<0.001
Number of admissions (mean, SD)	1.4 (2.1)	2.2 (2.3)	0.038
Home oxygen therapy (*n*, %)			0.834
Yes	238 (22.7)	8 (24.2)	
No	808 (77.3)	25 (75.8)	
BODEx (mean, SD)	2.7 (1.9)	3.4 (1.9)	0.099
BDI score (mean, SD)	8.8 (6.1)	17.2 (10.1)	<0.001
MMSE (mean, SD)	26.3 (4.0)	24.7 (5.6)	0.038
Charlson Index (mean, SD)	1.4 (1.4)	2.4 (2.4)	<0.001
CAT score (mean, SD)	21.5 (8.2)	26.3 (8.2)	<0.001
EQ-5D VAS (mean, SD)	57.7 (18.4)	48.3 (24.2)	0.006
EQ-5D tariff (mean, SD)	62.6 (24.1)	38.7 (27.5)	<0.001
Minutes walked per day (mean, SD)	67.1 (57.5)	37.4 (25.4)	0.007

Totals may not add up to the total number of patients due to missing values. Percentages are calculated without missing values. Comparisons were performed with the Fischer’s exact test for qualitative variables and the Student’s *t*-test for quantitative variables, Bonferroni-corrected. Abbreviations: SD, standard deviation; COPD, chronic obstructive pulmonary disease; BODEx, Body mass index, airflow Obstruction, Dyspnea and Exacerbations; BDI, Beck Depression Inventory; MMSE, Mini Mental State Examination; CAT, COPD Assessment Test; EQ-5D, EuroQoL-5 dimensions; VAS, visual analogic scale.

**Table 3 jcm-11-02558-t003:** Multivariate associations with suicidal ideation.

MODEL 1: Multivariate Analysis Including All The Variables	Odds Ratio	95%CI	*p*-Value
Sex (women vs men)	2.722	1.771–4.183	<0.001
BDI score	1.163	1.127–1.200	<0.001
Charlson Index	1.228	1.082–1.394	0.001
**MODEL 2: Multivariate Analysis Excluding Questionnaires**	**Odds Ratio**	**CI 95%**	** *p* ** **-Value**
Sex (women vs. men)	2.537	1.713–3.757	<0.001
Dyspnea grade	1.885	1.550–2.291	<0.001

Abbreviations: CI, confidence interval; BDI, Beck Depression Inventory.

**Table 4 jcm-11-02558-t004:** Multivariate associations with suicide attempts.

MODEL 1: Multivariate Analysis Including All The Variables	Odds Ratio	95%CI	*p*-Value
Age	0.945	0.912–0.980	0.001
BDI Score	1.160	1.112–1.210	<0.001
**MODEL 2: Multivariate Analysis Excluding Questionnaires**	**Odds ratio**	**CI 95%**	** *p* ** **-value**
Age	0.945	0.912–0.980	0.002
Number of outpatient exacerbations	1.213	1.087–1.354	<0.001

Abbreviations: CI, confidence interval; BDI, Beck Depression Inventory.

## Data Availability

Data are available from the authors upon request.

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
