# Peer review of "Frequency and Associated Factors of Suicidal Ideation in Patients with Chronic Obstructive Pulmonary Disease"

_jcm, 2022, doi:10.3390/jcm11092558_

Round 1
Reviewer 1 Report
This is an interesting article with a nice pool of patients from multiple centres across the country who were analysed.
The authors started off really well in describing the primary objectives and secondary objectives with a good methodology. However, after the results section, the discussion seemed a little haphazard and difficult to follow due to poor paragraphing. The discussion section and conclusion is probably the poorest part of the entire article. The authors should frame only ONE main point in each paragraph followed by supporting ideas and evidences /comparisons from other literatures. There should be also a better flow in the presentation of ideas across to the general reader. The authors have also failed to emphasise what is novel in their findings.
The conclusion seemed to have missed the objectives altogether and lack any emphasis on how their findings are novel and will change clinical practice.
It is a pity that anxiety and panic attacks were not included in the study as these are extremely common among COPD patients and would be of great clinical use if an association was found.
There are still minor grammatical errors in the article that needs to be corrected.
Author Response
This is an interesting article with a nice pool of patients from multiple centres across the country who were analysed.
The authors started off really well in describing the primary objectives and secondary objectives with a good methodology. However, after the results section, the discussion seemed a little haphazard and difficult to follow due to poor paragraphing. The discussion section and conclusion is probably the poorest part of the entire article. The authors should frame only ONE main point in each paragraph followed by supporting ideas and evidences /comparisons from other literatures. There should be also a better flow in the presentation of ideas across to the general reader. The authors have also failed to emphasise what is novel in their findings.
Response: We have rewritten some paragraphs of the Discussion following the suggestions of the reviewer and included some new references (refs. 27, 34 and 55).
The conclusion seemed to have missed the objectives altogether and lack any emphasis on how their findings are novel and will change clinical practice.
Response: We have included a new paragraph in the Discussion, at the end of the limitations section, highlighting the main findings of the study and their clinical implications.
It is a pity that anxiety and panic attacks were not included in the study as these are extremely common among COPD patients and would be of great clinical use if an association was found.
Response: We agree with the reviewer, this is a limitation that has been described in the limitations section of the Discussion.
There are still minor grammatical errors in the article that needs to be corrected.
Response: The text has been extensively reviewed.
Reviewer 2 Report
The manuscript is well written. The introductory section is nicely structured and includes the rationale and background for the results.
Unfortunately, the results seem to be at least incomplete (not to say manipulated). For that reason, I've decided to not give yet my comments to the discussion section.
Regarding results:
In line 116 authors declare that "1190 patients were valid for the analysis". Meanwhile, in Table 1 in line "variables" there is total of 1150 individuals (1046 "no" + 144 "yes") and in Table 2 there is total of 1146 "variables" (1112 "no" + 34 "yes").
Unfortunately, it's the tip of the iceberg.
Back to the Table 1. The total number of individuals who declared "no" in question regarding suicidal ideation was 1046, but only 1039 have sex (854 males + 185 females), only 1043 were included in the "coexistence" factor, only 1038 have educational level, only 1012 were included in the "smoking habit" factor. The total number of individuals who declared "yes" in question regarding suicidal ideation was 144, but only 143 have sex (94 males + 49 females), only 142 were included in the "coexistence" factor, only 142 have educational level, only 139 were included in the "smoking habit" factor.
Back to the Table 2. The total number of individuals who declared "no" in question regarding suicidal attempts was 1112, but only 1105 have sex (894 males + 211 females), only 1109 were included in the "coexistence" factor, only 1103 have educational level, only 1081 were included in the "smoking habit" factor. The total number of individuals who declared "yes" in question regarding suicidal attempts was 34, but only 33 have sex (24 males + 9 females), only 33 were included in the "coexistence" factor, only 33 have educational level, only 33 were included in the "smoking habit" factor.
The authors appear to have intentionally removed some answers or participants. Especially since the discrepancy in numbers has not been explained in the methods section.
Author Response
Response: We thank the reviewer for the comprehensive revision of our data. We want to make clear that the results have not been manipulated in any way. This is a very large database and some variables have missing values. This is the reason why the totals do not always add up to the total number of patients included. There is no adjudication of missing values. We apology for not being clear in the description of the data analysis.
We have now explained this in the text (Statistical analysis) and in the footnotes of the tables.
Round 2
Reviewer 2 Report
Dear Authors,
I absolutely realize how time-consuming and requiring is an analysis of such a large database. Even more so, every effort should be made to reliably describe and analyze such data. I also know that missing values may occur in almost every stage of the study. As researchers, it is our responsibility to reliably report such cases, and in most cases to reject incomplete questionnaires or surveys. Therefore, your response to my doubts “This is a very large database and some variables have missing values. This is the reason why the totals do not always add up to the total number of patients included.” is insufficient. Please provide sufficient information in the “statistical analysis” section as has been declared in response: “We have now explained this in the text (Statistical analysis) and in the footnotes of the tables.” There is only a meager explanation in the footnotes of the tables. The information regarding how many surveys have missing values, whether there was more than one missing value in individual surveys, and finally why they were not excluded from the study should be provided.
The authors' explanation did not dispel my doubts as to the reliability of the presented data. Since the authors declare that “We want to make clear that the results have not been manipulated in any way.“ and I assume that they are completely convinced of the correctness of the hypotheses put forward, I listed some of my observations and concerns below. Based on the presented data, I must say that the authors drew incorrect conclusions or presented the data illegibly.
- Methods
Taking into account the statement from chapter 2.2. : “Information about patients’ sociodemographics and clinical data were collected from face-to-face interviews with the patients and from medical records.” it is quite hard to believe that authors do not collect information regarding the sex of every single participant. In spite of that, it somehow occurred. It is absolutely necessary to point out how many surveys have missing values for each parameters measured as well as how many and ways were excluded from the study. Perhaps it will be clearly presented in the table. It is also important to give information on how many experimenters collect the data and is the same experimenters collect and analyze data?
2.3. Statistical analysis
Statistical analysis has been briefly and to laconically described e. g. sentence: “In order to determine the relationship between quantitative variables by group, the Student’s T test or the ANOVA tests were used, as appropriate.” Please provide information regarding the particular analysis performed to the description of the tables. It will definitely increase the reliability of the data.
- Results
The conclusion “As shown in Table 1, suicidal ideation was more frequent in women,” is inconsistent with the data presented. As we can read in table 1 among 144 individuals with previous suicidal ideation there were 49 females and 94 males so undoubtedly suicidal ideation was more (almost 2 times) frequent in men than in women. This needs to be explained. After much reflection and analysis of the data, I come to the conclusion that perhaps the authors concluded that more women commit suicide on the basis that, according to the reviewer's estimates, about 26% (N = 49) of all surveyed women (N = 234) had previous thoughts of suicide, while among all the men surveyed (N = 948) it was only 11% (N = 94). Then the application would be justified. However, the data presented in this manner as currently in the manuscript do not indicate this. It does not change the fact that this type of inference is unjustified, if only due to the significant difference in the number of men vs women. The results presented in Table 1 show many contradictions, apart from the abundance problem I mentioned earlier. It is not logical how the percentages in this table were calculated. Taking into account the first parameter mentioned: gender among people without prior suicidal thoughts (N = 1046), the authors declare that there are 82.2% of men (N = 854), while in fact there are 81.6% of them (X = (854 x 100%) / 1046) and 17.8% of women (N = 185), while in fact there are 17.7% of them (X = (185 x 100%) / 1046). The values ​​given by the authors add up nicely to 100% (82.2% + 17.8%) while in fact the amount is 99.3% (81.6% + 17.7%). I would also like to point out that the authors did not manage to match the percentages for the coexistence parameter nicely, and here the sum of these values for people with previous suicidal ideation is 99.9%. Analyzing and estimating the obtained data so that they look nice in the publication is manipulation. I absolutely do not understand why the authors did not choose to make it clear that there are missing values.
There is also the issue of the smoking habit to be clarified. There are only 2 groups on the tables, ex-smoker, and active smokers. What about people who have never smoked? Initially, I assumed that these are the missing values in the abundance for this parameter. However, in response to my earlier review, the authors explained that the lower numbers are due to the missing values. Please explain why the experiments were carried out only on people who have or had a smoking habit? Smoking what substance do the authors mean?
By reading the manuscript further we can find out that: “In those receiving domiciliary oxygen suicidal ideation was also more prevalent.” Meanwhile, in table 1 we can read that among 144 individuals with previous suicidal ideation there were 48 on home oxygen therapy and 94 without it so suicidal ideation was more prevalent in individuals NOT receiving domiciliary oxygen. This discrepancy needs to be explained.
The authors declare that “When patients were categorized into four groups according to BDI score (no depression, mild depression, moderate depression, and severe depression), frequencies of suicidal ideation were 2.9%, 4.9%, 13.3% and 36.5%, respectively (p < 0.001). “ However there is no explanation of how many individuals were in rach BDI score group. Is mentioned frequency a mean, median, or some other value? Provide missing information.
In lines, 128 to 139 authors describe some statistically significant results presented in table 1 and ignore others such as “coexistence”, “educational level” and others. A description of these data should be provided.
Since the role of the reviewer is not to convert the data for authors, let me not add comments to Table 2, while hoping that the authors will check the reliability of the data on their own and correct their description based on my comments in Table 1.
- Discussion
The text itself is written quite swiftly and logically. There are few sentences to comment on the differences between with and without previous suicidal ideation in aspects of coexistence and education level.
Author Response
Dear Authors,
I absolutely realize how time-consuming and requiring is an analysis of such a large database. Even more so, every effort should be made to reliably describe and analyze such data. I also know that missing values may occur in almost every stage of the study. As researchers, it is our responsibility to reliably report such cases, and in most cases to reject incomplete questionnaires or surveys. Therefore, your response to my doubts “This is a very large database and some variables have missing values. This is the reason why the totals do not always add up to the total number of patients included.” is insufficient. Please provide sufficient information in the “statistical analysis” section as has been declared in response: “We have now explained this in the text (Statistical analysis) and in the footnotes of the tables.” There is only a meager explanation in the footnotes of the tables. The information regarding how many surveys have missing values, whether there was more than one missing value in individual surveys, and finally why they were not excluded from the study should be provided.
Response: We already indicated in the Method section that patients with missing values in the key outcomes were excluded from the study (Method, page 2), but missing values in other variables were not a reason to exclude a patient from the study. Figure 1 indicates that 63 patients were excluded due to missing values in the outcome variables. The number of missing values in each variable can be easily obtained by simple calculations from the data in each table.
The authors' explanation did not dispel my doubts as to the reliability of the presented data. Since the authors declare that “We want to make clear that the results have not been manipulated in any way.“ and I assume that they are completely convinced of the correctness of the hypotheses put forward, I listed some of my observations and concerns below. Based on the presented data, I must say that the authors drew incorrect conclusions or presented the data illegibly.
Response: We are sorry to see that the reviewer thinks that our conclusions are incorrect or our data illegible. In fact, reviewer 1 had not concerns about our data and neither had the reviewers of our previous publications derived from the same database. Below, you will find our response to the concerns of the reviewer.
- Methods
Taking into account the statement from chapter 2.2. : “Information about patients’ sociodemographics and clinical data were collected from face-to-face interviews with the patients and from medical records.” it is quite hard to believe that authors do not collect information regarding the sex of every single participant. In spite of that, it somehow occurred. It is absolutely necessary to point out how many surveys have missing values for each parameters measured as well as how many and ways were excluded from the study. Perhaps it will be clearly presented in the table. It is also important to give information on how many experimenters collect the data and is the same experimenters collect and analyze data?
Response: As you can see in the Acknowledgement section there were 343 investigators recruiting at least one patient in the study, and sex was missing in only 8 cases out of 1190 (0.6%). It is possible that in these cases, investigators just failed to tick the box in the CRF corresponding to sex and this created a missing value. This type of mistakes is infrequent (0.6%) but may happen. In any case, these patients were not excluded from the study, they were only excluded from the multivariate analysis, because only cases without missing values in the dependent variables were included in this type of analysis.
The investigators that collected the date were different from the scientific committee that analysed the data. This has been included in Method (page 3)
2.3. Statistical analysis
Statistical analysis has been briefly and to laconically described e. g. sentence: “In order to determine the relationship between quantitative variables by group, the Student’s T test or the ANOVA tests were used, as appropriate.” Please provide information regarding the particular analysis performed to the description of the tables. It will definitely increase the reliability of the data.
Response: The point raised by the reviewer is well taken. All comparisons in the current analysis were conducted between 2 independent groups, and, therefore, only the Students’ T test was used. We have clarified this in the Methods section and have included a comment in the footnotes of the Tables.
- Results
The conclusion “As shown in Table 1, suicidal ideation was more frequent in women,” is inconsistent with the data presented. As we can read in table 1 among 144 individuals with previous suicidal ideation there were 49 females and 94 males so undoubtedly suicidal ideation was more (almost 2 times) frequent in men than in women. This needs to be explained. After much reflection and analysis of the data, I come to the conclusion that perhaps the authors concluded that more women commit suicide on the basis that, according to the reviewer's estimates, about 26% (N = 49) of all surveyed women (N = 234) had previous thoughts of suicide, while among all the men surveyed (N = 948) it was only 11% (N = 94). Then the application would be justified. However, the data presented in this manner as currently in the manuscript do not indicate this. It does not change the fact that this type of inference is unjustified, if only due to the significant difference in the number of men vs women. The results presented in Table 1 show many contradictions, apart from the abundance problem I mentioned earlier. It is not logical how the percentages in this table were calculated. Taking into account the first parameter mentioned: gender among people without prior suicidal thoughts (N = 1046), the authors declare that there are 82.2% of men (N = 854), while in fact there are 81.6% of them (X = (854 x 100%) / 1046) and 17.8% of women (N = 185), while in fact there are 17.7% of them (X = (185 x 100%) / 1046). The values ​​given by the authors add up nicely to 100% (82.2% + 17.8%) while in fact the amount is 99.3% (81.6% + 17.7%). I would also like to point out that the authors did not manage to match the percentages for the coexistence parameter nicely, and here the sum of these values for people with previous suicidal ideation is 99.9%. Analyzing and estimating the obtained data so that they look nice in the publication is manipulation. I absolutely do not understand why the authors did not choose to make it clear that there are missing values.
Response: We respectfully disagree with the reviewer. There is no doubt that suicidal ideation was more frequent in women. As the review mentions, suicidal ideation was present in 26% of women and only in11% of men (p<0.001), this is a fact, not speculation.
Regarding percentages, as the reviewer has correctly pointed out, there were some (n=8) missings in sex, therefore the denominator is not 1046, but 1038. We can’t include those patients with sex unknown in the denominator. As a consequence, among patients with known sex there were 82.2% of men and 17.8% of women, adding up to 100%, as presented in the table. We believe that this is the right way to present the data. This is not manipulation, all data are clearly presented.
There is also the issue of the smoking habit to be clarified. There are only 2 groups on the tables, ex-smoker, and active smokers. What about people who have never smoked? Initially, I assumed that these are the missing values in the abundance for this parameter. However, in response to my earlier review, the authors explained that the lower numbers are due to the missing values. Please explain why the experiments were carried out only on people who have or had a smoking habit? Smoking what substance do the authors mean?
Response: Smoking refers to cigarette smoking, as in all studies in COPD. There were only 2 groups: smokers and exsmokers, because, by exclusion criteria, all never smokers were excluded from the study. We apologise because this exclusion criterion was not described in the current manuscript. The detailed inclusion/exclusion criteria were described in previous manuscripts derived from this study. We have now included this criterion in the Method section (page 2)
By reading the manuscript further we can find out that: “In those receiving domiciliary oxygen suicidal ideation was also more prevalent.” Meanwhile, in table 1 we can read that among 144 individuals with previous suicidal ideation there were 48 on home oxygen therapy and 94 without it so suicidal ideation was more prevalent in individuals NOT receiving domiciliary oxygen. This discrepancy needs to be explained.
Response: Similarly to the analysis with sex, 33.8% of patients with home oxygen had suicidal ideation compared to 21.6% of those without (p=0.002). There is no doubt that suicidal ideation was more frequent in patient with home oxygen, as indicated in the text.
The authors declare that “When patients were categorized into four groups according to BDI score (no depression, mild depression, moderate depression, and severe depression), frequencies of suicidal ideation were 2.9%, 4.9%, 13.3% and 36.5%, respectively (p < 0.001). “ However there is no explanation of how many individuals were in rach BDI score group. Is mentioned frequency a mean, median, or some other value? Provide missing information.
Response: Apologies for this missing piece of information, the numbers are: No depression 310, mild depression 272, moderate depression 430, and severe depression 178. This information has now been included in the Results section (page 4). We have also included the same type of data for suicidal attempts on page 6.
In lines, 128 to 139 authors describe some statistically significant results presented in table 1 and ignore others such as “coexistence”, “educational level” and others. A description of these data should be provided.
Response: We only describe in the text the results that we find more relevant from the Table. One of the rules of writing a scientific paper is not duplicate in the text what is already presented in the Tables.
Since the role of the reviewer is not to convert the data for authors, let me not add comments to Table 2, while hoping that the authors will check the reliability of the data on their own and correct their description based on my comments in Table 1.
Response: The same comments apply to data presented in Table 2.
- Discussion
The text itself is written quite swiftly and logically. There are few sentences to comment on the differences between with and without previous suicidal ideation in aspects of coexistence and education level.
Response: We thank the reviewer for his/her comments.